# Strategy and Metrological Support for Indoor Radon Measurements Using Popular Low-Cost Active Monitors with High and Low Sensitivity

**DOI:** 10.3390/s24154764

**Published:** 2024-07-23

**Authors:** Andrey Tsapalov, Konstantin Kovler, Peter Bossew

**Affiliations:** 1National Building Research Institute, Faculty of Civil and Environmental Engineering, Technion, Israel Institute of Technology, Haifa 3200003, Israel; 2Retired, 1090 Vienna, Austria

**Keywords:** indoor radon, temporal variation, continuous radon monitors, sensitivity, calibration, metrology, measurement uncertainty, conformity assessment, decision-making, QA/QC

## Abstract

Traditionally, for indoor radon testing, predominantly passive measurements have been used, typically applying the solid-state alpha track-etch method for long-term and the charcoal method for short-term measurements. However, increasingly, affordable consumer-grade active monitors have become available in the last few years, which can generate a concentration time series of an almost arbitrary duration. Firstly, we argue that consumer-grade monitors can well be used for quality-assured indoor radon assessment and consequent reliable decisions. Secondly, we discuss the requirements of quality assurance, which actually allow for reliable decision-making. In particular, as part of a rational strategy, we discuss how to interpret measurement results from low-cost active monitors with high and low sensitivity with respect to deciding on conformity with reference levels that are the annual average concentration of indoor radon. Rigorous analysis shows that temporal variations in radon are a major component of the uncertainty in decision-making, the reliability of which is practically independent of monitor sensitivity. Manufacturers of low-cost radon monitors already provide sufficient reliability and quality of calibration for their devices, which can be used by both professional inspectors and the general public. Therefore, within the suggested measurement strategy and metrologically assured criteria, we only propose to clarify the set and values of the key metrological characteristics of radon monitors as well as to upgrade user-friendly online tools. By implementing clear metrological requirements as well as the rational measurement strategy for the reliable conformity assessment of a room (building) with radon safety requirements, we anticipate significant reductions in testing costs, increased accessibility, and enhanced quality assurance and control (QA/QC) in indoor radon measurements.

## 1. Introduction

### 1.1. Indoor Radon: Risk and Regulation

Radon is one of the most dangerous pollutants in indoor environments, exposure to which increases the likelihood of lung cancer as its concentration in buildings rises. Therefore, the World Health Organization (WHO) [1] and the International Commission on Radiological Protection (ICRP) [2] recommend, while the International Atomic Energy Agency (IAEA) [3], the European Basic Safety Standards (EU-BSS) [4], and the new Brazilian Basic Radioprotection Requirements [5] require, the implementation of a national reference level (RL) of 300 Bq/m^3^, limiting the *annual average radon concentration* in buildings. In EU legislation (binding to all Member States), this applies to residences and workplaces alike. RLs vary due to differences in regional radon levels and usually range from 100 to 300 Bq/m^3^ for dwellings [6] and up to 600–1000 Bq/m^3^ for workplaces with low occupancy. For example, the residential RL = 300 Bq/m^3^ in Germany, France, Spain, Portugal, and Brazil; RL = 200 Bq/m^3^ in the UK, Sweden, Italy, Canada, and Israel; and RL = 100 Bq/m^3^ in Denmark, Norway, and the Netherlands [6]. In the US, the indoor radon concentration is regulated through an Action Level (AL), set at 4 pCi/L (148 Bq/m^3^) [7,8], which is not equivalent to RL [9] and is discussed in Section 2.1. 

### 1.2. Indoor Radon Survey Strategies and Consumer-Grade Active Monitors

Historically, two different indoor radon survey strategies have emerged and are used, which serve the overall objective of reducing the health detriment caused by radon exposure. Their detailed discussion would exceed the scope of this paper, which is focused on technical–metrological aspects; very briefly, the approaches are as follows.

The first, which we may call the US approach [7,8], builds on as many individual indoor radon measurements as possible to identify buildings that require mitigation or remediation. The target is a decision on whether a building complies with the RL/AL or not. This strategy has led to an impressive number of measurements and mitigation/remediation actions [10]. 

The second strategy is laid down in the EU-BSS [4] and aims to implement a radon policy through regulation. A central tool identifies areas where radon exposure can be assumed high on average and prioritizes action there, leading to the concept of radon priority areas (RPAs), but attention to individual indoor testing is significantly weakened [9]. However, both the US and EU approaches assert that measuring indoor radon is necessary [11,12,13], and consequently, proper metrological quality assurance and quality control (QA/QC) are essential, to which the rest of this paper is devoted. 

The motivation behind this work is the widespread availability of quite cheap consumer-grade active radon monitors, which may revolutionize individual radon measurement practices, with the potential to also influence the regulatory approach of the EU. In fact, over the past few years, low-cost active radon monitors, priced around USD 150–350, have emerged in the indoor radon devices market and have already become popular, including for community measurement (which also has a certain appeal as a form of Citizen Science). These monitors are produced by companies such as Airthings, Ecosense, and FTLab. Tens of thousands of these low-cost (non-professional) radon monitors are sold annually, mainly in the US and EU countries. Our estimate suggests that approximately 0.5 to 1 million radon monitors have already been sold, and this number continues to increase. Additionally, our extensive experience [9], along with the research findings of other colleagues [14,15,16,17,18], indicates that low-cost radon monitors provide measurements of indoor radon that are sufficiently reliable (accurate and precise) in the context of radon abatement policy. Hence, there is already a vast number of reliable and affordable radon monitors that could be utilized by homeowners to test not only their homes but also neighboring residences, covering millions of buildings across different countries. However, an unsolved problem remains regarding ensuring high-quality radon measurements, which can be carried out en masse by the population themselves (without the participation of professionals) to assess the conformity of their homes and offices with radon safety requirements.

### 1.3. Quality Assurance in Indoor Radon Measurement

However, neither the US standards (ANSI/AARST MAH-2023 [7], MA-MFLB-2023 [8]) nor the international standards (ISO 11665-8 [19]) currently propose a rational procedure for indoor radon measurements based on a clear principle and mathematical algorithm [20], which would include a metrologically supported quantitative criterion for the effective and reliable conformity assessment of a room with a norm. Specifically, one must distinguish between *performing an individual measurement* of indoor radon concentrations and its *interpretation with regard to a target*, which is currently not covered by the harmonious international standard with strict metrology. In particular, this pertains to the decision as to whether the RL is exceeded or not.

More generally, QA must always be understood as a QA chain [21], which leads from strict metrological support for indoor radon measurements to reliable decision making via defined procedures. After all, the target of radon policy—which evidently includes indoor radon measurement—is to achieve correct decisions, that is, decisions which lead to a reduction in radon exposure individually and collectively. The steps to achieve this target include correctly interpreting the measurement results in relation to the target. 

The unclear situation regarding the scope of QA seems to be the reason why manufacturers of both low-cost and professional radon monitors attempt to independently offer (through their websites and online platforms) what they consider to be the most effective ways for displaying and analyzing the results of continuous indoor radon measurements. However, the proposals from monitor manufacturers, like the standards mentioned above, traditionally rely on qualitative algorithms and criteria and lack a rigorous scientific (metrological) foundation. 

There is a commonly held belief (though not entirely shared by the authors) that a high efficiency of QA/QC in indoor radon testing can be achieved primarily through the high sensitivity of radon monitors that ensure accurate and precise measurements of radon concentration and fast response times. However, it is important to understand that the main objective of indoor radon measurement is not just to measure radon concentrations with a certain (controlled) accuracy and precision or to monitor radon dynamics in a tested room. In fact, the main objective is to assess the conformity of a room or a building with safety requirements (identification of potential hazards) by making a reliable decision on whether an RL that limits the *annual average radon concentration* is exceeded or not. Reliable means that the second-kind or false-negative decision error probability is less than 5% (second-kind error means that an effect is not detected although in reality it exists).

### 1.4. The Temporal Aspect of Indoor Radon Measurements

It is especially important to emphasize that the reliability of decision-making regarding compliance (or non-compliance) with regulatory requirements depends less on the accuracy and precision of radon concentration measurements (expressed as the instrumental uncertainty) but rather on the temporal uncertainty arising from significant variations in indoor radon concentrations over time, as shown in the article [20]. This paper also presents a statistical approach that allows for the simultaneous consideration of all natural and anthropogenic factors affecting temporal uncertainty so that the traditional investigation of the influence of each factor separately on the temporal behavior of indoor radon becomes unnecessary.

Unfortunately, the key role of temporal uncertainty in decision-making has yet to receive sufficient attention and quantitative consideration by both national regulators and the radon measurement community. This is further evidenced by the fact that in recent years, numerous articles [14,15,16,17,18] have been published dedicated to the examination of components and factors influencing the instrumental uncertainty of low-cost radon monitors. Furthermore, aside from our work, no one else has attempted to investigate the temporal (key) uncertainty of indoor radon [20]. For example, recently published studies that explore the seasonal (temporal) radon concentration correction factor [22] or the relationship between short- and long-term measurements [23] or the influence of various factors on indoor radon behavior [24,25] do not contain quantitative data on temporal uncertainty. Therefore, the findings of articles [22,23,24,25] as well as other researchers cannot be used for the purpose of the harmonization and improvement of QA/QC in indoor radon measurements within rational ISO/IEC concepts such as ‘measurement uncertainty’ [26] and ‘conformity assessment’ [27]. Although these concepts were established 10–20 years ago and were adopted in measurement standardization globally with the support of the Joint Committee for Guides in Metrology [28], the recommendations of such authoritative bodies have yet to be integrated into the practice of indoor radon regulation due to the entrenched conservatism of national regulators [20,29,30].

One might ask, why not simply increase the measurement duration as much as possible to significantly reduce temporal uncertainty so that it does not need to be taken into account when making decisions? The established conservative approach in Europe advocates precisely such a solution. The reason is that radon levels are low (well below the RL) in the vast majority of buildings, so it is waste of time and other resources to measure low *annual average radon concentrations* with high accuracy in every building if the objective is to decide compliance with the RL. In addition, long tests are not always possible or convenient; for house transactions, one needs a result within a few days. Also, for the routine assessment of individual houses, users may not be inclined to wait for the result of a one-year measurement (which is the norm in German radon surveys, for example). Therefore, metrological quality assurance including the temporal component must be available. 

Taking all this into account, the goal of our research is to determine the contributions of the main components of temporal and instrumental uncertainties to the uncertainty budget of an estimate of the *annual average radon concentration*, considering the duration of measurements and the sensitivity level of low-cost radon monitors from various manufacturers. The results of such computational research within a rational criterion for conformity assessment [20] enable us to justify and propose a strict (quantitative) and metrologically supported algorithm that provides the most effective (operational) strategy for indoor radon measurements to identify rooms (buildings) where the *annual average radon concentration* does not exceed the RL with a probability of at least 95%. Additional algorithms for the rational identification of hazardous rooms (buildings) are discussed. Recommendations to manufacturers of radon monitors to clearly express target metrological parameters and improve online tools for indoor testing are also discussed.

## 2. Uncertainty Budget of Annual Average Radon Concentration

### 2.1. Rational Criterion for Conformity Assessment

Within ISO/IEC concepts [26,27], the core of a rational conformity assessment criterion is to compare the RL with the upper bound of the confidence interval for the estimated annual average radon concentration. This is determined on the basis of the measurement results of the indoor radon concentration, considering the possibility of controlling the main components of combined uncertainty (instrumental and temporal), which decrease with increasing the measurement duration. The rational criterion for assessing the conformity of a room with a normative at a given (manageable) reliability of decision making (at least 95%, no more than 5% false-negative error) for both short- and long-term measurements is expressed as follows [20,29,30]:(1)$$C\left(t\right)\cdot \left[1+\sqrt{{U_{V}(t)}^{2}+{U_{D}}^{2}}\right]<C_{RL}$$
where *C_RL_* is the reference (normative) level for the annual average indoor radon concentration, Bq/m^3^.

*C*(*t*) is the measured (mean) radon concentration (Bq/m^3^) over the measurement period of *t* (in hours), which is determined as follows [20]:(2)$$C\left(t\right)=\frac{n_{g}/t-n_{0}/t_{0}}{CF}=\frac{r_{g}-r_{0}}{CF}\;$$
where *n_g_* and *r_g_* are the number of counted pulses and the count rate (in cph) of the gross effect obtained during the measurement period *t*.

*n*_0_ and *r*_0_ are the number of counted pulses and the count rate (in cph) of the background effect obtained during the background measurement period *t*_0_, according to the instructions of the monitor manufacturer.

*CF* is the calibration factor (within ISO-accepted terminology) or monitor sensitivity (within terminology used by device manufacturers), expressed by the net count rate per Bq/m^3^ or cph/(Bq/m^3^); this parameter can be determined using (2) in an experiment in which a “true” (established by other means) value of *C*(*t*) is known.

*U_V_*(*t*) is the temporal uncertainty of indoor radon, which is defined and discussed in the separate Section 2.2 due to the preeminent importance of this component (here and throughout the text, the uncertainty is given in relative units, for example, 0.25 or 25%, while uncertainty expressed in % is not used in formulas; in addition to temporal uncertainty, the same applies to the expression of uncertainty of other parameters mentioned below).

*U_D_* is the instrumental (device) uncertainty that combines all sources of uncertainty (mainly random/statistical and systematic/calibration components) associated with the measured radon concentration regardless of the nature of radon origin and behavior of radon in time and space, which is determined as follows [20]:(3)$$U_{D}=k\cdot \sqrt{\frac{r_{g}/t_{g}+r_{0}/t_{0}}{(r_{g}-r_{0}{)}^{2}}+{u(CF)}^{2}\;}$$
where *k* is the coverage factor (here, equal to 2) used as a multiplier of the standard uncertainty in order to obtain an expanded uncertainty that defines an interval with an increasing level of confidence, approximately from 68% at *k* = 1 to 95% at *k* = 2 or up to 99% at *k* = 3 in the case of a normal distribution [28].

*u*(*CF*) is the CF uncertainty with *k* = 1. 

Considering (2) and (3) and assuming *r*_0_ = 0, Criterion (1) can be expressed as follows:(4)$$C(t)\cdot \left[1+\sqrt{U_{V}(t{)}^{2}+{U_{CF}}^{2}+\frac{k^{2}}{C(t)\cdot CF\cdot t}}\right]<C_{RL},$$
where *U_C__F_* is the CF uncertainty with *k* = 2 discussed in detail in Section 2.3, and the last term under the square root in (4) is associated with the statistical uncertainty *U_St_*(*t*), which also depends on the measurement duration and is expressed as:(5)$$U_{St}(t)=\frac{k}{\sqrt{C(t)\cdot CF\cdot t}}$$

By the way, as mentioned in Section 1.1, it is useful to highlight that AL (adopted in the US regulation) and RL are different parameters [9], linked by Criterion (1), where the maximum value of *C*(*t*) represents AL. However, in the US standards (clause 7.1 in ANSI/AARST MAH-2023 [7] and MA-MFLB-2023 [8]), the parameters AL and RL are compared directly, and it is mistakenly assumed that AL and RL have the same meaning.

### 2.2. The Temporal Uncertainty of Indoor Radon

The assessment of temporal uncertainty *U_V_*(*t*) is based on the accumulation and processing of year-long time series of indoor radon concentrations measured in a representative sample of buildings [9,20,30]. The idea is a statistical analysis of the deviations of indoor radon concentrations from annual average levels, measured under diverse conditions in a large number of houses. 

More formally, *U_V_*(*t*) is defined as the value of the 95-th percentile (or 95% probability) in the distribution of all relative deviations *D_ij_*(*t*) between the measured concentrations *C_ij_*(*t*) and the measured annual average concentration (AAC) in the *j*-th room, *C_j_^AAC^* [30]: *D_ij_*(*t*) = [*C_j_^AAC^*/*C_ij_*(*t*)] − 1,
where index *j* refers to the monitored rooms (houses) or time series, numbered from 1 to *N*, so *j* = 1…*N*, and index *i* counts the number of data (measured concentrations) from 1 to *M*, so *i* = 1…*M*, contained in each time series. In each *N* house (one monitored rooms in each house), year-long continuous measurements (YLCMs) with a registration/integration period of 1 or 3 h are performed. Thus, initial monitoring is an array or time series that includes either *M* = 8760 (at *t’* = 1 h) or *M* = 2920 (at *t’* = 3 h) measurement data.

According to the original algorithm [20], the initial monitoring data can be transformed into a new time series with the same value *M* and any value *t* greater than the initial measurement duration (period) *t’*. This approach makes it possible to obtain from one initial time series, for example, 20–30 additional time series (for the same monitored room) with different measurement durations *t*, the set of values of which covers the target interval from 1 to 360 days. It is clear that *U_V_* (*t* = 1 year) = 0. 

For each *t*, the frequency distribution of *D_ij_*(*t*) is built, consisting of *N* × *M*(*t*) values and the 95% quantile estimated, which is the temporal uncertainty *U_V_*(*t*). The higher the number *N* of time series included, the better the representativeness of the houses (rooms) monitored, and the more reliable the resulting statistic of *D_ij_*(*t*). 

According to [9,20,30], addressing this problem requires conducting YLCMs with 1 or 3 h registration periods in about *N* = 200 to 300 houses of different types with elevated radon levels within an international or national case study. Conducting YLCMs under diverse conditions and in many houses of different designs and located in different geological/climatic conditions allows for obtaining a statistically representative array for verifying and improving conservative values of *U_V_*(*t*), as presented in Table 1. This straightforward approach allows for statistically accounting for the cumulative influence of all anthropogenic and natural factors on temporal uncertainty, considering the test duration. As the number of YLCMs included in the analysis increases, taking into account influencing factors such as climate, geology, and room and building characteristics, the temporal uncertainty can most probably be reduced.

### 2.3. Uncertainty of the Calibration Factor 

The CF uncertainty constitutes a systematic component of instrumental uncertainty. In the context of the industrial production and metrological certification of radon monitors, CF uncertainty mainly includes: (i) bias error (indicating lack of accuracy) and (ii) standard deviation (lack of precision) that characterizes the spread of real individual sensitivity values between monitors of the same model. Additional factors such as fluctuations in indoor air aerosol concentration, humidity, and temperature can contribute to an increase in CF uncertainty. Moreover, CF uncertainty should take into account the non-linearity of calibration across the operational measurement range. Thus, it is advisable to conduct calibration and assess CF uncertainty with varying radon concentrations over time so that environmental conditions inside buildings, typically ranging from around 10 to 500–1000 Bq/m^3^, are adequately represented. This range covers the principal target (reference, action, or other control) levels of indoor radon, while higher indoor concentrations such as 1000–10,000 Bq/m^3^ are rare. Moreover, measurements of very high concentrations do not require high accuracy within criteria (1) and (9). Consequently, any room with natural radon behavior at an average radon concentration between 50 and 500–1000 Bq/m^3^ can be used for metrological tests, considering that the lower the concentration, the longer the test duration, according to (5). 

Under these considerations, CF uncertainty (at *k* = 2) is defined as 2 × square root of the sum of the squares of the following two components:(a)*relative biased error* = (*measured mean*/*true value*) − 1;(b)*relative standard deviation* = *standard deviation*/*measured mean*.
where *true value* denotes the average activity concentration of radon during the test measured with a reference instrument, and *measured mean* and *standard deviation* represent the calculated results from the simultaneously measured activity concentration of radon during the test using 20–30 monitors of the same model with factory calibration.

The duration of such a metrological test should ensure a statistical uncertainty of no more than 0.05 (*k* = 2) for both the reference and tested monitors, so that its influence can be disregarded. Note that the calculated CF uncertainty (at *k* = 2) may be greater than or equal to the CF uncertainty (at *k* = 2) of the reference monitor. In the case of verification of an individual monitor, only component (a) is taken into account, and the calculated CF uncertainty (at *k* = 2) also cannot be less than the CF uncertainty (at *k* = 2) of the reference monitor. Otherwise, the calculated CF uncertainty (at *k* = 2) should be taken as equal to the CF uncertainty (at *k* = 2) of the reference monitor.

### 2.4. Summary of Uncertainty Budget Components

According to the expression under the root in (4), the uncertainty budget of the annual average radon concentration includes the three following relative contributions—*ρ_V_*, *ρ_CF_* and *ρ_St_* (%):-From the temporal uncertainty:
(6)$$\rho_{V}=100\cdot \frac{U_{V}(t{)}^{2}}{U_{V}(t{)}^{2}+{U_{CF}}^{2}+{U_{St}(t)}^{2}}$$-From the CF uncertainty:
(7)$$\rho_{CF}=100\cdot \frac{{U_{CF}}^{2}}{U_{V}(t{)}^{2}+{U_{CF}}^{2}+{U_{St}(t)}^{2}}$$
-From the statistical uncertainty:
(8)$$\rho_{St}=100\cdot \frac{{U_{St}(t)}^{2}}{U_{V}(t{)}^{2}+{U_{CF}}^{2}+{U_{St}(t)}^{2}}$$



## 3. Characteristics of Low-Cost Radon Monitors

Table 2 compares the main technical characteristics of popular low-cost (non-professional) radon monitors. 

The important metrological characteristics such as the CF uncertainty and device background are not provided by the manufacturers in Table 2. Additionally, the calculated measurement duration associated with the given statistical uncertainty vary significantly. These facts confirm the absence of unified and clear metrological requirements and algorithms available both to device manufacturers and to national metrological services responsible for regulating indoor radon. The rational (effective and metrologically supported) verification of low-cost radon monitors is discussed below.

## 4. Results and Discussion

### 4.1. Composition of the Combined Uncertainty Budget 

Based on the detailed mathematical framework presented in Section 2 and using Equations (5)–(8) and data from Table 1, a computational investigation was carried out to determine the contributions of the main components as temporal and instrumental (CF and statistical) uncertainties to the combined uncertainty budget of the annual average radon concentration, taking into account the measurement duration and the sensitivity of low-cost radon monitors from different manufacturers, using the data from Table 2. The most important results are presented in the following paragraphs.

From Figure 1, it can be observed that the contribution of statistical uncertainty is so small that this component does not affect the decision-making reliability, according to Criterion (1), when measurements are conducted with highly sensitive monitors (see Table 2). At the same time, the contribution of CF uncertainty increases with increasing the measurement duration due to a decrease in both temporal uncertainty, according to Table 1, and statistical uncertainty, according to (5). However, the contribution of CF uncertainty to the combined uncertainty becomes significant only for measurement durations exceeding 2–3 months, and the value of *U_CF_* does not exceed 0.4 (*k* = 2). It is also important to highlight that the contribution of temporal uncertainty *U_V_*(*t*) is predominant if measurement durations are not more than 6, 7, or 8 months with *U_CF_* = 0.4, 0.3, or 0.2, respectively. 

It may seem surprising, but approximately the same observations and conclusions (as in the previous paragraph) apply to monitors with low sensitivity, with the only difference being that at low indoor radon concentrations (less than 50 Bq/m^3^), the contribution of statistical uncertainty is more pronounced, according to Figure 2. However, even in this case, the negligible contribution of statistical uncertainty has almost no impact on decision-making, especially at indoor radon concentrations at or above the reference levels (*C_RL_*) of 100 Bq/m^3^ or higher.

An important conclusion emerges from the analysis of the data in Figure 1 and Figure 2; the reliability of decision-making based on Criterion (1) is practically independent of the sensitivity level of the radon monitors presented in Table 2. Another significant finding is that the CF uncertainty can be assumed to be 0.4 (*k* = 2) for both high- and low-sensitivity monitors without noticeably decreasing the decision-making reliability, provided that the duration of continuous measurements does not exceed 2–3 months.

Furthermore, and this is the main conclusion, we can clearly see that the temporal component is the most important source of annual average concentration uncertainty for measurement durations of up to 6 months. This is the main motivation as to why the study of temporal uncertainty is the most urgent task in modern radon metrology, without solving which it is impossible to improve QA/QC in indoor radon measurements. 

### 4.2. Assessment of Calibration Uncertainty

The results of the study [15] show that among 20 RadonEye Plus2 monitors, the CF uncertainty is about 0.20–0.25 (*k* = 2), using data in figures rather than tables and texts. We can assume that manufacturers (FTlab and Ecosense) may intentionally and reasonably underestimate the CF value (2) by 5–15%, which leads to the same slight increase in the measured radon concentrations in the range up to 500 Bq/m^3^, which is most commonly found in buildings. Indeed, the underestimation of the CF value does not lead to a decrease in decision-making reliability (i.e., hazard identification). Considering this and the conclusions of the previous subsection, there is no need to adjust the CF value (sensitivity of monitors set by the manufacturer) or to apply an individual correction factor, which was assessed in [15] as well as in [16,17].

In the study [17], tests were conducted on 36 RadonEye Plus2 monitors, the results of which suggest that the CF value of monitors of this model is indeed a little underestimated; however, the CF uncertainty does not exceed 0.2 in the range of radon concentrations from 2000 to 7000 Bq/m^3^, also considering the non-linearity of the calibration over the measurement range. Unfortunately, the results of this study do not allow us to estimate the CF uncertainty in the range from 20 to 500 Bq/m^3^. Later, the same authors provided updated data [16], from which it can be concluded that the average CF value among 36 RadonEye Plus2 monitors is underestimated by 1/0.873 = 1.145 times or by 14–15% in the range below 3500 Bq/m^3^. Meanwhile, the CF uncertainty is approximately 0.25 with *k* equal to 2 or even 3. If the bias in the CF value of RadonEye Plus2 monitors is eliminated, the CF uncertainty will be significantly less than 0.25. It is also worth noting that another part of the article [16] is dedicated to studying the RadonEye’s response time, although this parameter does not influence the reliability of decision-making within the framework of rational criteria (1) and (9), including a measurement strategy, the algorithms of which are discussed in detail in the sections below.

In another study [18], tests of low-cost monitors were conducted over four rounds at different radon concentrations (around 200, 400, and 1000 Bq/m^3^), as well as at elevated temperature (+30 °C) and humidity (70%) in one of the rounds with 200 Bq/m^3^. Each round lasted 7 days, which ensured a negligible contribution of the statistical component in the above radon levels, so that the statistical uncertainty can be ignored when assessing CF uncertainty. In summary, the CF uncertainty regarding three RadonEye Plus and three Wave Plus monitors did not exceed 0.2 (*k* = 2) and 0.3 (*k* = 2), respectively.

In the study [14], the following *individual* values of CF uncertainty for different low-cost monitors over a test period of about 8 (and 2) weeks under natural (extreme) variations in radon over time are provided (the statistical component’s contribution can also be neglected): 0.09 (0.06) for RadonEye Plus2, 0.16 (0.13) for RadonEye, 0.05 (0.31) for EcoQube, and <0.11 (<0.40) for low-sensitivity Airthings monitors (Table 2).

Regarding the high-sensitivity EcoQube and low-sensitivity Airthings radon monitors, it would be useful to conduct additional research to refine the CF uncertainty (at *k* = 2) using 20–30 devices of each monitor model according to the guidelines in Section 2.3. However, summarizing the above CF uncertainty data from different studies [14,15,16,17,18] in combination with our experience of using 17 RadonEye Plus2 and RadonEye monitors over many years, it can be argued that the manufacturers of low-cost radon monitors mentioned in Table 2 already provide sufficient reliability and quality of calibration for their monitors (CF uncertainty does not exceed 0.30), which can be used not only by residents and building owners but also by professional inspectors. 

In this regard, it would be highly beneficial for developers of national and international standards to consult directly with leading manufacturers of radon monitors, taking into account their opinions and valuable experience when developing QA/QC standards in indoor radon measurements. Here, we also clearly see that underestimation and even a neglect of the contribution (key role) of temporal uncertainty leads to unreasonably high requirements for monitor accuracy in [15,16,17], such as with the assessment of an individual correction factor and other non-target parameters, which significantly complicates indoor radon metrology for low-cost monitors.

Given the high level of current QA/QC among radon device manufacturers, to avoid excessive requirements for metrological support of measurements using radon monitors, we recommend that manufacturers publish data on the main metrological characteristics of monitors in the specifications for each model, such as sensitivity (CF) and CF uncertainty (at *k* = 2), as well as the validity period of these metrological data (3 or 5 years). 

### 4.3. Verification of Monitor Sensitivity

Upon expiration of the validity period of metrological data specified by manufacturers, residents (owners of low-cost monitors) or professional inspectors could periodically independently verify (confirm or reject) the monitor sensitivity by parallel measurements with another monitor whose validity period of metrological characteristics has not expired. To conduct such a verification, a large box or room with a radon concentration (regardless of its temporal variations) at a mean value from 50 to 500–1000 Bq/m^3^ is required. The statistical uncertainty of each monitor should be <0.05 (at *k* = 2) over the test period, the duration of which is determined by Equation (5). The criterion of satisfactory sensitivity of the tested monitor can be the difference in the measured radon concentrations not exceeding, for example, its CF uncertainty (at *k* = 2). Otherwise, further operation of the tested monitor should be discontinued.

When a radon monitor with a CF uncertainty of about 0.20 (at *k* = 2) is verified, testing should be conducted at a national (accredited) metrology service using a more accurate reference monitor with a CF uncertainty of <0.20 (at *k* = 2). Only in this case may it be necessary to use an individual correction factor.

### 4.4. The Problem of Background Control 

An important additional metrological characteristic of a radon monitor is its background, which is usually expressed through parameters such as *n*_0_ or *r*_0_ or such as the equivalent radon concentration, according to (2), where instead of the difference (in the numerator), either *n*_0_ or *r*_0_ is simply substituted. An analysis of published data [15] shows that among 20 RadonEye Plus2 monitors, the average background value is 2.5 Bq/m^3^ with a standard deviation of 0.5 Bq/m^3^. In another study [17], the background of nine RadonEye Plus2 monitors ranged from 2.2 to 3.8 Bq/m^3^. These data seem to be quite relevant, and such a low background level firstly indicates that the detectors are functioning properly, and secondly, it practically does not affect the measurement results, even at low indoor radon concentrations of around 20 Bq/m^3^. At the same time, in the study [14], a significantly higher background of 17 Bq/m^3^ is reported for a RadonEye Plus2 monitor. For other models of low-cost monitors studied in in [14], a similarly high background is mostly reported, ranging from 11 to 20 Bq/m^3^, which likely indicates a significant influence of factors that were not considered when planning the study to measure monitor backgrounds. In this regard, it would be useful to conduct additional research to identify factors that contribute to background overestimations and then develop a reliable method for determining the background of radon monitors. 

In this context, we recommend that manufacturers of radon monitors publish (in the specifications) data on the background of each monitor model in the form of the maximum background radon concentration value that cannot be exceeded among new monitors (e.g., <5 Bq/m^3^). For monitors that have been used for long time, the maximum background may be higher (e.g., <10 Bq/m^3^); it is therefore advisable to also control the background through parallel measurements in outdoor air or in a well-ventilated room using a new monitor. The duration of such a test should ensure a statistical uncertainty <0.10 (at *k* = 2) for each of the monitors, according to (5). A satisfactory check of the background of an operating monitor can be determined by a difference in measured radon concentrations not exceeding, for example, 5 Bq/m^3^, considering the monitor background limit levels (10 − 5 = 5 Bq/m^3^) suggested above. Otherwise, further operation of the tested monitor should be discontinued.

It is important to note that an elevated monitor background does not lead to a reduction in the reliability of decision-making aimed at hazard identification based on a rational Criterion (1), considering also that the RL is significantly higher than 10 Bq/m^3^. Therefore, the background of a radon monitor may not be considered in the mathematical algorithm of the rational criterion, namely in Formulas (2)–(5). However, background checking is recommended every 3–5 years of operation of radon monitors, or more frequently if the measured radon concentration is consistently above 1000 Bq/m^3^.

### 4.5. Rational Measurement Strategy

The radon concentrations in over 90% of buildings are significantly lower than the reference level (in countries with elevated indoor radon, the RL is also usually higher), rendering long-term measurements of low radon concentrations unnecessary at every location. In most cases, a reliable decision can be made through a 4–6-day test within Criterion (1) if the representativeness of *U_V_*(*t*) is ensured. In this scenario, radon monitor owners can quickly test rooms in their homes (within 1–3 months), usually confirming the absence of a radon problem. Consequently, the continued use of the monitor in their households becomes irrelevant. Nevertheless, it is useful to continue using the monitor to test neighboring homes and offices where the monitor owner’s relatives, friends, and acquaintances spend significant time. Note that in a relatively small number of cases (mainly in radon priority areas), longer measurements (several months or a whole year) guided by Criterion (1) may be necessary.

To demonstrate the online operation of the rational Criterion (1), Figure 3 presents a year-long time series of radon concentrations measured by RadonEye Plus2, transformed to make the annual average concentration (AAC) equal to 50 Bq/m^3^ for ease of analysis. In addition, the dynamics of the measured mean concentration and of the calculated maximum AAC are shown, considering the increasing duration of continuous measurements. The maximum AAC is the calculated value of the upper bound of the confidence interval for the AAC (with 95% probability), which is online compared to the RL, according to (4). If the calculated maximum AAC does not exceed *C_RL_* = 100 Bq/m^3^ during 4–6-day measurements, as seen in Figure 3A, it can be concluded that the AAC in the tested room does not exceed *C_RL_* = 100 Bq/m^3^ with 95% probability, and the test can be completed. It should be noted that the data in Figure 3 are independent of the monitor’s sensitivity, and there is no visible difference between the calculated maximum AAC at *U_CF_* = 0.20 (at *k* = 2) and 0.40 (at *k* = 2), if the measurement duration is not more than 2–3 months, as discussed in detail above. At the same time, for long-term indoor testing, there is no need to use monitors with *U_CF_* < 0.20 (at *k* = 2) due to the existence of year-to-year variations in indoor radon [32], which can serve as a benchmark for the maximum achievable accuracy within Criterion (1).

When discussing the example in Figure 3A, it should be noted that measurements could have started, for instance, 2 months later when the radon concentration in the same room significantly increased, as observed in Figure 3B. In this scenario, the maximum AAC drops below *C_RL_* = 100 Bq/m^3^ only after 4 months of measurements, which cannot be considered a quick decision about the compliance of a room with a norm. However, even in this case, as depicted in Figure 3B, if *C_RL_* = 150 or 200 Bq/m^3^ (i.e., three or four times higher than AAC), a quick decision about the room’s compliance with a norm can still be made within 4–6-day measurements. According to UNSCEAR reports, the radon concentration in the vast majority of buildings ranges from 30 to 50 Bq/m^3^, while the RLs are usually in the order of 200 or 300 Bq/m^3^. Therefore, in most cases, 4–6-day measurements are sufficient for quickly assessing the compliance of a room with a norm.

The time series in Figure 3 is just a special case of thousands of indoor radon behavior options. It is important to understand that regardless of the nature of the temporal variations in radon and the moment when testing has started, the reliability of the results that consists of comparing a calculated maximum AAC with an RL (or making a decision about the compliance of a tested room with radon safety requirements) will be at least 95% (for test duration > 2 days) if the representativeness of *U_V_*(*t*) is ensured. Note that in 5% of cases, or 20 out of 365 days a year, the start of testing may occur at very low radon levels and cause a false-negative decision error. To reduce this error, indoor radon testing must be carried out in the normal mode of room usage, that is, when windows are not constantly open or closed for a long time. Temporal uncertainty for the closed mode is significantly less [20,30], but it has been studied even less compared to the normal mode. This is the main motivation as to why the study of temporal uncertainty is an urgent task of modern metrology for indoor radon measurements, as already reported above, including in Section 2.2.

In the relatively rare cases where the maximum AAC does not drop below the RL within 1–2 weeks of measurements, i.e., Criterion (1) is not met, the following actions can be recommended:

(i) Simultaneously with Criterion (1), additionally utilize Criterion (9) for identifying hazardous rooms, which is discussed in the subsection below; if Criterion (9) is not met either, then either:

(ii) Continue active measurements for an indefinite duration (up to a whole year), waiting for the maximum AAC to drop below RL, or

(iii) Utilize the SSNTD method for measurements lasting 9–12 months, which provides good accuracy for long-term measurements (*U_D_* < 0.15 at *k* = 2) and also has a reasonable cost of approximately USD 45, according to the website www.radonova.com.

Option (ii) is more suitable for residents or non-professional users. However, before embarking on prolonged measurements with indefinite duration, it is recommended to conduct a short-term test for 1–2 weeks in other residential or office rooms with high and long-term occupancy to determine the optimal strategy for further long-term measurements, and, if necessary, for example, purchase additional radon monitors.

Option (iii) is more suitable for professional inspectors since it is economically disadvantageous to use even low-cost (non-professional) radon monitors for measurements that take 9–12 months. Furthermore, such a monitor should have a *U_CF_* of about 0.20 (at *k* = 2), according to the data in Figure 3, which requires additional significant expenses for monitor verification by the national metrological service.

Within such a rational measurement strategy, the cost-effectiveness of using low-cost radon monitors (priced at USD 150–350 regardless of sensitivity) with a *U_CF_* of from 0.30 to 0.40 (at *k* = 2) and a service life of 3–5 years lies in the low cost of radon testing, ranging from USD 1 to 2, if the measurement duration is 1–2 weeks. This is much cheaper than even short-term measurements by the charcoal method.

To implement this opportunity, we recommend that radon monitor manufacturers develop user-friendly online tools that automatically enable science-based decision-making using Criteria (1) and (9) and their options. Standardization of measurements on a rational basis will legitimize public involvement and the use of low-cost monitors in regulating indoor radon. This would empower the public to independently conduct indoor testing by low-cost radon monitors with higher efficiency and reliability compared to existing professional testing practices when the temporal (key) uncertainty in decision-making is not taken into account. The approach we propose will increase radon tests by providing convenient online tools and reducing costs, particularly crucial in low-income areas where testing is limited. Additionally, the rational strategy will enhance the efficiency and reliability of indoor testing in real estate transactions and assessing mitigation effectiveness.

### 4.6. Additional Criterion for Identifying Rooms with High Radon

Within the proposed rational measurement strategy, it is advantageous to introduce, in addition to Criterion (1), an additional criterion for identifying hazardous rooms in which the RL may be exceeded with a specified probability (95%, 98%, or 99%). The additional criterion for identifying hazardous rooms (buildings) at a given (manageable) reliability of decision-making (at least 95%) for both short- and long-term measurements is expressed as follows:(9)$$\frac{C\left(t\right)}{MF(t)}\cdot \left[1-\sqrt{{U_{CF}}^{2}+\frac{k^{2}}{C\left(t\right)\cdot CF\cdot t}}\right]>C_{RL}\;$$
where *MF*(*t*) is the multiplicity factor defined as the value of 95-th percentile (or 95% probability) in the distribution of all ratios between the measured concentrations *C_ij_*(*t*) and the measured AAC: *DEV_ij_*(*t*) = *C_ij_*(*t*)/*C_j_^AAC^* (*i* = 1…*M*; *j* = 1…*N*), in a representative sample of *N* buildings (rooms) within an international or national case study. It is important to clarify that the arrays of values *C_ij_*(*t*) and *C_j_^AAC^* are the same [20,30] that were used to determine the temporal uncertainty of indoor radon *U_V_*(*t*) (see Section 2.2). The *MF*(*t*) values are provided in Figure 4.

It is also useful to specify that in Criterion (9), the contribution of statistical uncertainty (the second term under the square root) is negligible and can be disregarded, especially in conditions where the measured radon concentration exceeds the RL by more than 1.5 times (Figure 4) and the test duration is at least 4–6 days regardless of the sensitivity of the radon monitor.

Thus, the application of Criterion (9) in conjunction with Criterion (1) allows for the identification of rooms (buildings) where the RL is either exceeded if Criterion (9) is met or not exceeded if Criterion (1) is met with a given reliability (no less than 95%). In relatively rare cases where both Criteria (1) and (9) are not met, it is necessary to continue measurements according to recommendations (ii) and (iii) in the preceding subsection.

### 4.7. Collection of Measurement Results

The collection of radon measurement results is an important national regulatory task that allows for the refinement of collective risk assessments and the delineation of radon priority areas (RPAs). In this regard, in addition to the location, time period, and result of measurement (*C*(*t*) ± *U_D_* in Bq/m^3^ at *k* = 2), it would be useful to additionally interview the occupiers and record the following minimum characteristics of the test conditions as metadata: (a) type of room and building, (b) floor level, (c) number of floors, (d) period of construction of the building, (e) building material, (f) type of ventilation and heating, and (g) ventilation mode—normal or closed [20]. 

An additional characteristic of the measurement conditions can be the name of the model of the radon monitor used as well as the values of its metrological characteristics, such as CF uncertainty and device background. If necessary, taking this data into account can increase the accuracy of collective risk assessments as well as RPA parameters.

Collection of data is easy to organize if the radon monitor used is controlled through the manufacturer’s online platform if data protection is guaranteed. In addition, online mapping of measurement data in a form more easily understandable to the public [33] than traditional maps [34] could serve as an important tool for informing the population about radon hazards. For example, the design of the map by the RadonTest Group [35] offers a detailed and understandable representation of the spatial distribution of measured indoor radon concentrations for the public while maintaining the confidentiality of personal data. At the same time, test participants can easily locate their measurement results on the map.

### 4.8. Recommendations to Manufacturers of Radon Monitors

The combination of the above considerations and intermediate conclusions allows us to formulate a comprehensive set of recommendations for manufacturers of radon monitors regarding the clear expression of target metrological parameters and the improvement of online tools for indoor testing. It is recommended that manufacturers of both professional and low-cost radon monitors publish data in the specifications for each monitor model in the form of the target metrological parameters mentioned in Table 2:(a)Sensitivity (or calibration factor),(b)CF uncertainty at *k* = 2, using the term “calibration uncertainty” instead of “accuracy” in the specification,(c)Upper measurement range (the lower range depends on the sensitivity, CF and statistical uncertainties, measurement duration, and device background; therefore, it is preferable to indicate the device background instead of the lower measurement range, which has no practical significance in metrology for indoor radon measurements within Criteria (1) and (9)),(d)Device background as the maximum equivalent radon concentration (Bq/m^3^) acceptable for new and long-term-used monitors,(e)Calculated measurement duration (in hours or days) to achieve a statistical uncertainty of <0.10 at *k* = 2 (without considering the background) for a radon concentration of 100 Bq/m^3^,(f)Specification of the validity period of these metrological parameters (3 or 5 years).

Manufacturers of radon monitors are also recommended to upgrade user-friendly online tools that automatically enable science-based decision-making within a rational measurement strategy and the metrologically validated Criteria (1) and (9) by implementing the following options in a manual and mobile application:(a)Continuous indication and storage of measured radon concentration values with integration periods of 1 or 3 h for monitors with high sensitivity and 12 or 24 h for monitors with low sensitivity *without using the principle of moving averages*; outputting data every 10 min is unnecessary,(b)Continuous indication of the mean value of the radon concentration, *C*(*t*), after 3–6 h for monitors with high sensitivity and 1–2 days for monitors with low sensitivity after the start of the test (Figure 3), including the counting of the measurement duration in days,(c)Continuous indication of the instrumental uncertainty (*U_D_*) at *k* = 2 for both the current and mean radon concentrations,(d)Continuous indication of the current calculated (predicted) maximum annual average concentration of indoor radon (Figure 3) for comparison with the relevant reference level (RL) at the test location, beginning 2 days after its start,(e)Provision of relevant messages, including a recommendation for test completion if Criterion (1) or Criterion (9) is met after 4–6 days of testing,(f)Online display of recorded pulses or the mean pulse count rate (after the start of measurements), including the duration of measurements in hours and minutes, in a special mode, for example, “verification”,(g)Questionnaire for recording metadata, collecting test results (without personal data), and displaying them on a radon map with an updated design, according to Section 4.7,(h)It is also recommended to display the measured radon concentration even if the upper measurement range is exceeded, as there is no requirement to ensure accuracy at the level of the established CF uncertainty for radon concentrations exceeding 5000 or 10,000 Bq/m^3^.

## 5. Conclusions

1. Overall, for measurement durations of up to 6 months, the temporal uncertainty contributes the most to the combined uncertainty budget of the annual average radon concentration. This applies to the conformity assessment of a room (building) with radon safety requirements using the rational Criterion (1) and the measurement strategy based on it. Underestimating or even neglecting the contribution (key role) of temporal uncertainty leads to unreasonably high requirements and complications of metrology, and it also does not allow for the development of a rational strategy for indoor radon measurements.

2. The study results and interim conclusions rely on existing conservative values of the temporal (key) uncertainty of indoor radon, *U_V_*(*t*) (see Section 2.2), which were obtained from limited statistical data. Better values of *U_V_*(*t*) do not exist yet. Therefore, to ensure the representativeness of *U_V_*(*t*) it is necessary to conduct year-long continuous measurements of indoor radon concentrations in a large number of diverse buildings (at least 300) located in different countries with different climates and geologies. This is one of the most urgent tasks in modern radon metrology, without solving which it is not possible to improve QA/QC in indoor radon measurement.

3. The reliability of the decision regarding the non-exceedance of the reference level (RL) according to Criterion (1), or the exceedance of the RL based on Criterion (9), is practically independent of the sensitivity level of low-cost radon monitors presented in Table 2.

4. The uncertainty of calibration (CF uncertainty) for both high- and low-sensitivity monitors should not exceed 0.4 (*k* = 2) within Criterion (1) if the duration of continuous measurements is no more than 2–3 months. If the test is conducted by a professional inspector for more than 3 months, the CF uncertainty should be around 0.20 (*k* = 2), regardless of monitor sensitivity.

5. Manufacturers of low-cost radon monitors, mentioned in Table 2, already provide sufficient reliability and quality of calibration for their radon monitors (in fact, the CF uncertainty does not exceed 0.30), which can be used by professional inspectors and the public. The use of low-cost radon monitors under online control by the rational measurement strategy within Criteria (1) and (9) would allow the public to independently conduct indoor testing with higher efficiency and reliability compared to existing professional testing practices when the temporal (key) uncertainty in decision-making is not taken into account. 

6. The implementation of the proposed clear metrological algorithms with more lenient requirements for sensitivity and accuracy of active radon monitors, along with the rational measurement strategy, will improve the reliable conformity assessment of tested rooms with radon safety norms. This approach also includes a set of clear recommendations for manufacturers of radon monitors, which will significantly reduce testing costs, increase their widespread availability, and improve quality assurance and quality control (QA/QC) in indoor radon measurements. Furthermore, the rational strategy will enhance the efficiency and reliability of indoor testing in real estate transactions and in assessing the effectiveness of mitigation measures.

7. ViewPlus, ViewRadon, WaveRadon, WavePlus, and Corentium Home monitors have low sensitivity, while RadonEye Plus2, RadonEye, and EcoQube monitors are highly sensitive. However, RadonEye lacks a timer, and EcoQube loses about 20% of the data. Therefore, currently, RadonEye Plus2 seems to be the best monitor for studying the behavior of indoor (as well as outdoor) radon among the low-cost monitors presented in Table 2.

8. Internationally, we advocate for the integration of the extended quality assurance concept, specifically the QA chain, into standardization regimes. In many cases, the focus is less on accurate measurements of radon concentrations and more on the conclusions drawn from those measurements regarding the safety or potential danger of homes. 

## Figures and Tables

**Figure 1 sensors-24-04764-f001:**
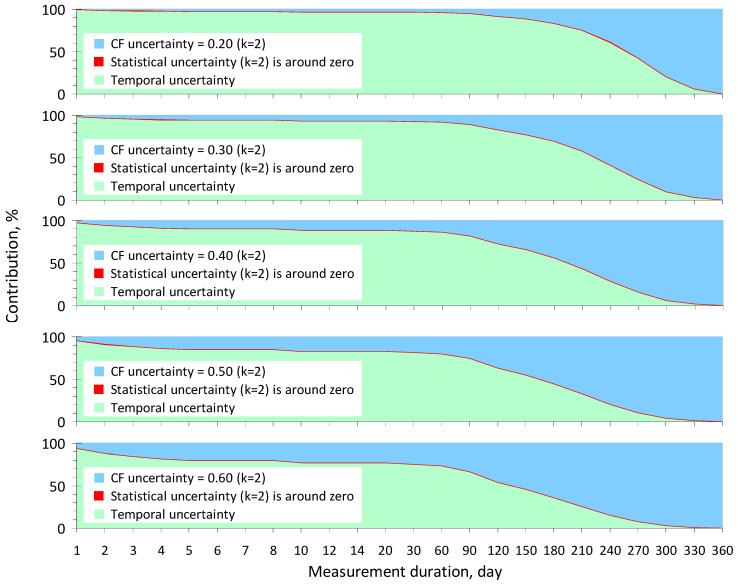
The contributions of temporal (Table 1), statistical (at *C*(*t*) > 10 Bq/m^3^), and CF (at *U_CF_* = 0.20, 0.30, 0.40, 0.50, and 0.60) uncertainties to the combined uncertainty budget of the annual average radon concentration depending on the duration of measurements for high-sensitivity monitors (Table 2).

**Figure 2 sensors-24-04764-f002:**
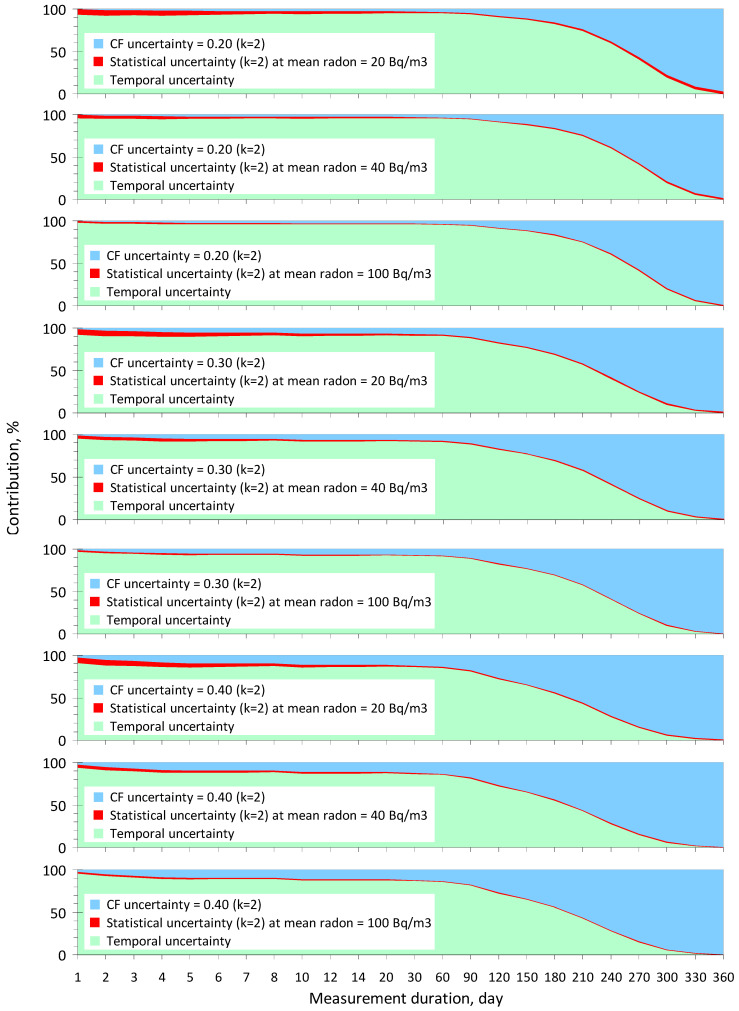
The contributions of temporal (Table 1), statistical (at *C*(*t*) = 20, 40, or 100 Bq/m^3^), and CF (at *U_CF_* = 0.20, 0.30, or 0.40) uncertainties to the combined uncertainty budget of the annual average radon concentration depending on the duration of measurements for low-sensitivity monitors (Table 2).

**Figure 3 sensors-24-04764-f003:**
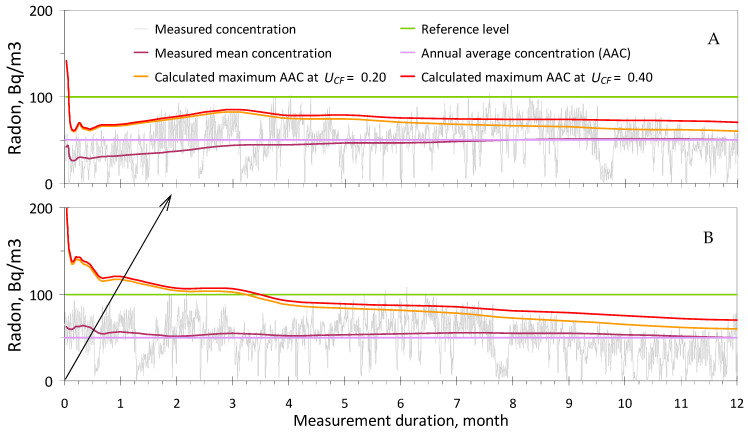
Example of the dynamics of radon, measured mean *C*(*t*), and calculated maximum annual average concentrations of indoor radon (at *U_CF_* = 0.20 and 0.40), using both high- and low-sensitivity monitors (Table 2) at an annual average radon concentration of 50 Bq/m^3^ ((**A**) initial time series starting with low radon concentrations, (**B**) the start of measurements (and calculation) is delayed by 2 months, beginning with elevated radon concentrations).

**Figure 4 sensors-24-04764-f004:**
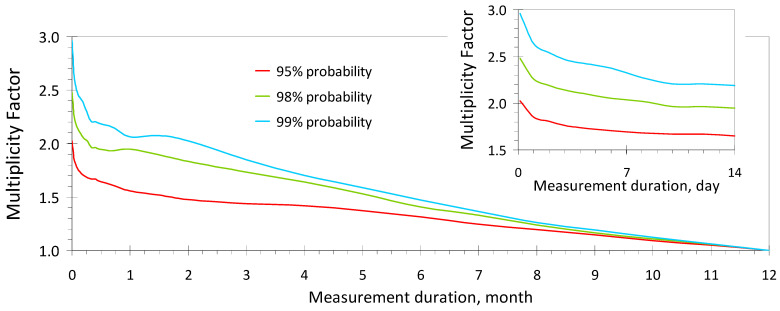
Multiplicity factor depending on the duration of measurements and the coverage probability in decision-making within Criterion (9).

**Table 1 sensors-24-04764-t001:** Conservative values of the temporal uncertainty depending on the measurement duration [9,30,31].

Day	1	2	3	4	5	6	7	8	10	12	14	20
*U_V_*(*t*)	2.30	1.60	1.40	1.25	1.20	1.20	1.20	1.20	1.10	1.10	1.10	1.10
Month	1	2	3	4	5	6	7	8	9	10	11	12
*U_V_*(*t*)	1.05	1.00	0.85	0.65	0.55	0.45	0.35	0.25	0.17	0.10	0.05	0.00

**Table 2 sensors-24-04764-t002:** Characteristics of low-cost radon monitors given by the manufacturers [14].

Manufacturer		FTLab (Republic of Korea)	Ecosense (USA)	Airthings (Norway)
Monitor model		RadonEye and RadonEye Plus2	EcoQube	View Radon, View Plus, Wave Plus, Wave Radon, and Corentium Home
Sensitivity level		High		Low
Sensitivity (or calibration factor) in cph at 1 Bq/m^3^/cpm at 1 kBq/m^3^		0.84/14		0.025 */0.42
CF uncertainty (*k* = 2)		no data		no data
Upper measurement range ** (Bq/m^3^)		3700 and 9435	3700	20,000
Device background as the maximum equivalent radon concentration (Bq/m^3^) acceptable for new/long-term used monitors		no data/no data		no data/no data
Statistical uncertainty	Manufacturer assessment	<10% at 370 Bq/m^3^ after 10 h		<10% at 200 Bq/m^3^ after 7 days
	Our assessment using (5)	<10% at 370 Bq/m^3^ after 1.5 (0.5) h at *k* = 2 (*k* = 1)		<10% at 200 Bq/m^3^ after 3.5 (1) day at *k* = 2 (*k* = 1)
		<10% at 100 Bq/m^3^ after 5 h at *k* = 2		<10% at 100 Bq/m^3^ after 7 days at *k* = 2
Data lost (%) [14]		2.2 and 2.4	18.9	0.8, 1.6, 6.1, no data for the last two models

* Data provided by the manufacturer following inquiry from the authors. ** The lower range depends on the sensitivity, the CF and statistical uncertainties, the measurement duration, and the device background, so it is better to indicate the device background instead of the lower measurement range.

## Data Availability

Data are contained within the article.

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
