# Peer review of "Strategy and Metrological Support for Indoor Radon Measurements Using Popular Low-Cost Active Monitors with High and Low Sensitivity"

_sensors, 2024, doi:10.3390/s24154764_

Round 1

Reviewer 1 Report

Comments and Suggestions for Authors

This paper presents a statistical study on evaluating the compliance of indoor radon concentrations measured with commercial instruments with safety standards. The study serves as a guide for both the general public and professionals in conducting indoor radon safety assessments. The paper aligns with the scope of the journal. However, there are certain issues to rectify before recommendations can be provided.

1. The layout of the article needs adjustments, and it is recommended that the textual narrative be placed close to the charts.

2. References should be numbered in the order in which they appear in the text.

3. It is recommended that the fonts in the article be standardized.

4. The formulas are now poorly formatted for display, and it is recommended that they be adjusted.

5. Initial letters need to be capitalized where abbreviations first appear. For example, in line 219, it should be Year Long Continuous Measurements (YLCM).

6. In line 191 and Eq.3, what is the physical meaning of k? An explanation should be added to the text.

7. What is the relationship between k and the CF uncertainty? How is it derived?

8. In Fig.1 and Fig.2, what physical model did you use in determining the individual component error contributions? Why do the contributions from CF increase as the measurement duration increases?

9. Line 635, remove f) in the text.

Comments on the Quality of English Language

The language can be improved.

Author Response

Please see the att. table.

Reviewer 2 Report

Comments and Suggestions for Authors

The paper titled”Strategy and Metrological Support for Indoor Radon Measurements Using Popular Low-Cost Active Monitors with High and Low Sensitivity” discuss the effects of some factors such as temporal variations in radon, monitor sensitivity, Calibration Factor uncertainty, and statistical uncertainty, on quality assured indoor radon assessment and consequent reliable decisions using consumer-grade active monitors. The paper is The subject of the study is interesting and current.

Part of the content in the conclusion, such as 6,7 (lines 616-650), would be better discussed in the Results and Discussion section.

Author Response

Please see the table for reviewer 1

Reviewer 3 Report

Comments and Suggestions for Authors

The authors of the paper deal with a current and important topic of radon metrology. They used data from long-term series of indoor radon measurements to determine the uncertainty contributions. The concepts and conclusions of the work are scientifically sound, but the presentation of the results needs improvements. 

(1) The nomenclature in the text, formulae, tables and figures is not consistent / harmonised and needs to be standardised carefully to increase readability. The same terms and formula symbols should be used in all the figures and tables as in the text.

(2) The references to the literature are helpful, but it is still necessary to briefly summarise the key facts and data in the text. It is insufficient to simply refer to the literature without including shortly and comprehensive the relevant content in the text of the paper.  

(3) It is necessary to specify the details of the data used in the details.

(4) Some of the given references are not accessible - please check all and correct if necessary. 

(5) Specify all relative uncertainties in % (text, tables, figures) 

(6) Consider carefully the comments in the attached PDF file.

(7) Improve the substructure in chapter 4 to improve the presentation and increase the readability.  

Author Response

Please see the table for reviewer 1

Round 2

Reviewer 1 Report

Comments and Suggestions for Authors

The authors addressed all my questions. 

Comments on the Quality of English Language

The English looks good to me.

Reviewer 3 Report

Comments and Suggestions for Authors

The authors have largely taken into account the objections I raised. In the case of the rejection of my objections and recommendations, this was sufficiently and comprehensibly argued. In my view, the revised version of the manuscript is in order and I have no further objections or recommendations.